# New Developments in the Pathogenesis, Therapeutic Targeting, and Treatment of Pediatric Medulloblastoma

**DOI:** 10.3390/cancers14092285

**Published:** 2022-05-03

**Authors:** Francia Y. Fang, Jared S. Rosenblum, Winson S. Ho, John D. Heiss

**Affiliations:** 1Department of Orthopedics, University of Maryland School of Medicine, Baltimore, MD 21201, USA; francia.fang@som.umaryland.edu; 2Neuro-Oncology Branch, National Cancer Institute, National Institutes of Health, Bethesda, MD 20892, USA; jared.rosenblum@nih.gov; 3Department of Neurosurgery, The University of Texas at Austin, Austin, TX 78712, USA; winson.ho@austin.utexas.edu; 4National Institute of Neurological Disorders and Stroke, National Institutes of Health, Bethesda, MD 20892, USA

**Keywords:** pediatric brain tumors, medulloblastoma, molecular subtype, chemotherapy, surgery, radiotherapy, imaging, cost of treatment

## Abstract

**Simple Summary:**

Medulloblastoma is the most common pediatric brain tumor, comprising one-third of all pediatric brain tumors, and originating in the posterior fossa of the brain. The disease is categorized into four subtypes: WNT, Sonic hedgehog (SHH), Group 3, and Group 4. Each subtype has unique pathogenesis, biomarkers, prognosis, response to therapy, and potential for further pharmacologic investigation. For example, it has recently been found that tumors in the SHH group arise due to aberrant persistence of defective cells of the embryonic germinal layer of the cerebellum. Herein, we review the recent critical advancements in understanding the four molecular subtypes that continue to shape our diagnostic, surgical, radiotherapeutic, and chemotherapeutic intervention and management to aid in treating pediatric medulloblastoma. We describe preclinical studies and clinical trials that aim to improve risk-stratification of disease, reduce therapy toxicity, and optimize the treatment of pediatric patients.

**Abstract:**

Pediatric medulloblastoma (MB) is the most common pediatric brain tumor with varying prognoses depending on the distinct molecular subtype. The four consensus subgroups are WNT, Sonic hedgehog (SHH), Group 3, and Group 4, which underpin the current 2021 WHO classification of MB. While the field of knowledge for treating this disease has significantly advanced over the past decade, a deeper understanding is still required to improve the clinical outcomes for pediatric patients, who are often vulnerable in ways that adult patients are not. Here, we discuss how recent insights into the pathogenesis of pediatric medulloblastoma have directed current and future research. This review highlights new developments in understanding the four molecular subtypes’ pathophysiology, epigenetics, and therapeutic targeting. In addition, we provide a focused discussion of recent developments in imaging, and in the surgery, chemotherapy, and radiotherapy of pediatric medulloblastoma. The article includes a brief explanation of healthcare costs associated with medulloblastoma treatment.

## 1. Introduction

Medulloblastoma is a central nervous system (CNS) tumor of cerebellar origin that comprises approximately 1% of all brain tumors [1,2]. However, medulloblastoma is the most common malignant brain cancer in children, accounting for 25–30% of childhood brain tumors and over 40% of posterior fossa childhood tumors [3]. Thus, medulloblastoma is primarily a childhood cancer, with an annual incidence of 300–350 new cases in the United States [4]. Most patients present before 16 years of age, with over 70% before 10, a third of which are younger than 3 years old; very few cases present under 1 year old [5,6]. The median age of diagnosis in children is about 5–7 years [7]. This age distribution highlights the current understanding of these tumors as remnants of aberrant embryonic cerebellar cells [8].

Classically, medulloblastoma has four molecular subtypes: WNT, sonic hedgehog (SHH), Group 3, and Group 4 [9]. Each subtype has unique molecular and clinical characteristics. WNT and SHH subtype medulloblastomas result from mutations in the WNT and SHH signaling pathways, respectively, while Groups 3 and 4 medulloblastoma etiologies require further elucidation. These groups have been well-studied, and continue to be clinically useful in prognostication and treatment, while also prompting further investigation which has led to further subclassification, including the association of P53 mutations with the SHH group in LFS syndrome, in the WHO 2021 guidelines [10]. Generally, the WNT subtype of medulloblastoma has the best prognosis, and the Group 3 subtype with p53 mutation has the worst [10,11]. Recent investigations into the pathogenesis of the SHH subtype medulloblastoma have provided great insight into the presentation and prognosis associated with this group, linking the clinical behavior of these tumors to the impact of the mutation on their development. Over the past decade, mutations in each subtype have been studied in relation to their pathogenesis. Understanding this variation provides insight into the clinical behavior of these tumors, and allows for the development of targeted therapies. SHH and WNT subtypes have benefited from the most remarkable advancements, while Group 3 and 4 subtypes trail far behind [12]. The subgroups continue to be further studied and elucidated by modern techniques including methylation, sequencing, and microarrays, which will inevitably lead to more detailed diagnostic schema. 

This review updates the understanding of the four subtypes of medulloblastoma, of treatment advances prolonging survival, and of treatment trials. Finally, we discuss the financial burden of medulloblastoma on patients, families, and the medical system. 

## 2. Clinical Overview

### 2.1. Characteristic and Presentation

Medulloblastoma typically arises within the posterior fossa in the cerebellum or its junction with the brainstem [13]. Recent studies have provided great insight into how the developmental biology of these tumors predicts their clinical behavior. For example, a recent lineage tracing study of the SHH subtype of medulloblastoma demonstrated that these tumors arise during the post-natal period from aberrantly persistent undifferentiated neural crest cells within the transient external germinal layer, which forms by embryonic migration of another transient structure, the rhombic lip. These cells ultimately become mature granule neurons within the internal granular cell layer of the cerebellum after the first year of life. These cells may not complete migration—establishing themselves anywhere along a path from the rhombic lip, which forms a critical interface between the cerebellum and brainstem in the embryo—into their proper position within the cerebellum. The tumor location critically affects the clinical course of these tumors in early childhood. A relationship between molecular pathogenesis, tumor location, and clinical behavior is seen not only in the SHH medulloblastoma subtype but also in the other medulloblastoma subtypes [14].

Typically, medulloblastoma arises within the medulla and expands the cerebellum to obstruct the fourth ventricle, creating signs of increased intracranial pressure such as major morning headaches, nausea, vomiting, and altered mental status. Compression of the adjacent brainstem structures and exiting cranial nerves may produce focal neurological deficits. For example, tumors within the cerebellar vermis and hemispheres cause gait ataxia and focal limb incoordination, respectively [15]. If left unchecked, tumor cells may disseminate through the cerebrospinal fluid (CSF) to the spinal canal. In rare instances, medulloblastoma may metastasize systemically to bone and bone marrow [16,17]. Tumors confined to the cerebellum have different symptoms than those metastasizing through the CSF and bloodstream.

### 2.2. Diagnosis

Medulloblastoma should be suspected clinically in children presenting with signs and symptoms of a posterior fossa mass, such as those listed above. Diagnosis requires brain imaging. The most accurate imaging modality is magnetic resonance imaging (MRI). Medulloblastoma typically appears hypointense and hyperintense relative to grey matter on T1- and T2-weighted MRI, respectively. Intravenous contrast injection produces contrast enhancement throughout the tumor on MRI or CT [18]. The location of the tumor on imaging reflects the developmental pathogenesis described above. The mass usually originates in the inferior vermis, and enlarges to compress the fourth ventricle. Depending on the molecular subgroup, the tumor may involve the cerebellar hemispheres, the cerebellopontine angle, or it may mimic meningioma by originating in the tentorium [19,20]. Medulloblastoma may spread to the spinal canal, forming leptomeningeal “drop metastasis” on the nerve roots of the cauda equina. Medulloblastoma may also originate in extra-axial locations [21,22,23,24]. The various molecular and histopathological subtypes of medulloblastoma have distinguishing MRI features (Figure 1) [25]. Diffusion-weighted imaging is characteristic of medulloblastoma compared to other posterior fossa tumors [26,27]. While radiographic imaging can hone in on the correct diagnosis, confirmation of medulloblastoma is made on hematoxylin and eosin staining of histopathologic specimens obtained during surgical resection, which is part of the standard treatment regimen as outlined below. Molecular techniques further subtype the medulloblastoma [9]. Surgical resection improves prognosis, although near-total resection (NTR) does not increase recurrent risk compared to gross total resection (GTR). It is not recommended to achieve GTR over NTR at the cost of neurologic deficits [3,28]. 

### 2.3. World Health Organization Classification and Grade

As discussed above, a panel of experts agreed at a conference held in Boston in 2010 to use molecular profiling to subdivide medulloblastoma into four different subtypes (WNT, SHH, Group 3, and Group 4) [13]. The molecular subtyping was separate from the 2016 WHO classification of medulloblastoma into four histologic subtypes: classic, desmoplastic/nodule, large cell/anaplastic, and medulloblastoma with extensive nodularity (MBEN). Most large cell/anaplastic medulloblastoma tumors belong to the SHH, Group 3, and Group 4 molecular subtypes, while the classic morphology was more typical of WNT tumors. Some SHH subtype tumors have desmoplastic/nodular morphology [29,30]. The updated 2021 World Health Organization (WHO) central nervous system tumor classification combined the four histologic subtypes into a single type, “Medulloblastoma, histologically defined.” The 2021 WHO classification further elaborated the medulloblastoma molecular subtypes: the SHH subgroup was divided into four groups, and the Group 3 and 4 subtypes (non-SHH, non-WNT) into eight groups [10]. These separations enabled more precise therapeutic targeting of the different medulloblastoma subtypes by conventional therapies and experimental therapies like gene therapy. As described below, each subtype group was defined by tumor genotyping, immunohistochemistry, and an individual prognosis. 

## 3. Molecular Subtyping

Molecular subtypes of medulloblastoma are characterized by unique molecular features and anatomic features, as described in Figure 2. Subgroup-specific treatments are summarized in Table 1. 

### 3.1. WNT

The WNT-pathway-derived subtype comprises 10% of medulloblastoma cases, and generally affects children over four years of age [13]. These tumors arise in the cerebellar midline and may spread to the dorsal brainstem [37]. They have the best prognosis of the four subtypes, with nearly 100% 5-year survival and rare metastasis [30,38,39,40]. Most WNT-derived tumors have a somatic CTNNB1 mutation, encoding B-catenin, and chromosomal variations like monosomy 6 [41,42]. Due to its favorable prognosis and low-risk biological profile, several clinical trials are investigating de-escalating treatment methods. Clinical trials NCT01878617, NCT02066220, and NCT02724579, examining reduced doses of craniospinal radiation and chemotherapy, are expected to report their results within the next few years [14]. Recent studies have found that WNT medulloblastoma secretes WNT antagonists that prevent blood-brain barrier formation. This aberrant tumor vessel phenotype permits chemotherapy to reach and achieve high concentration within the WNT-subtype tumor, contributing to its excellent prognosis [31]. Unlike the SHH subtype, recurrence of WNT subtype medulloblastoma is not associated with the extent of resection and metastatic disease [28,43,44]. Anatomically, most relapses occur in the lateral ventricles; the remainder occur in the leptomeninges and surgical tumor bed or, in rare cases, in the suprasellar region [45,46].

### 3.2. SHH

The SHH subtype is the most common medulloblastoma subtype, with an estimated 25–30% prevalence. The SHH subtype usually occurs in children less than 5 years old [38,47]. The tumors arise in the cerebellar hemispheres, and are characterized by mutated or inactivating mutations of PTCH1 (43%) and SUFU (10%) [37]. This subtype has an 80% overall survival in large cohort trials [39,48]. A multicenter phase 2 trial from St. Jude’s (SJYC07; NCT00602667) divided the SHH subtype into two molecular subgroups: iSHH-I and iSHH-II, based on their distinct methylation patterns. In patients under 5 years of age not receiving radiation, intraventricular chemotherapy, or high-dose chemotherapy, the iSHH-II group had improved progression-free survival compared to iSHH-I [49]. The prognosis of SHH subtype medulloblastoma further depends on TP53 gene status, the most critical risk factor for SHH medulloblastoma. While SHH is generally distributed in young children, SHH/TP53 mutant tumors occur almost exclusively in children older than 5 years, of whom 56% carry a TP53 germline mutation. The TP53 mutation confers a relatively poor prognosis, with a 41% five-year overall survival rate, compared to 81% in the SHH subtype without the TP53 mutation. This mutation is responsible for 72% of deaths in children with the SHH subtype [50]. Relapse of the SHH subtype medulloblastoma generally occurs in the surgical tumor bed [51]. The SHH pathway’s two main molecular targets are Smoothened (SMO) receptor antagonists and GL1 transcription factor inhibitors [52]. Small and large molecules cannot cross the blood-brain barrier of this subtype, making it resistant to chemotherapy, unlike WNT subtype tumors. The development of nanoparticles, which may encapsulate SMO or GLI inhibitors, may improve drug delivery [32,33]. Nanoparticle technology provides a promising new research avenue for treating SHH subtype medulloblastoma, based on testing in in vitro and in vivo medulloblastoma models. 

### 3.3. Group 3

Group 3 comprises approximately one-quarter of medulloblastoma cases, affecting infants and young children. Males predominate 2:1 to females in Group 3 [53]. Group 3 is an aggressive medulloblastoma subtype, with over 40% of patients having metastases at diagnosis. The five-year survival rate ranges between 30% and 60% [12,14,54]. Like WNT-type tumors, Group 3 tumors grow in the midline and often into the fourth ventricle. The Group 3 subtype tumor recurs almost exclusively in the leptomeninges [51]. No therapeutic agents effectively target Group 3 medulloblastoma or reduce its high mortality. A better understanding of its underlying tumor molecular genetics and molecular pathways will provide drug targets that can overcome its therapeutic resistance. Genotyping and transcriptomics combined with methylation have successfully identified many biomarkers that may direct future therapies. About 20% of Group 3 tumors have *MYC* oncogene amplification, which carries a poor prognosis [55]. Transcriptome and tissue studies have also identified follistatin-like 5 (FSTL5) immunopositivity as a poor prognostic factor in Group 3 and Group 4 tumors [56]. Other molecular identifiers in Group 3 include increased expression of *GFI*, *IMPG2*, *GABRA5*, *EGFL11*, *NRL*, *MAB21 L2* [57,58], and *MYC-*driven long non-coding RNAs (lncRNAs). These genes driving Group 3 medulloblastoma are being targeted in preclinical animal models [57]. 

Group 3 tumor growth depends on angiogenesis, as evidenced by the significant elevation of *VEGFA* mRNA expression in this group compared to other medulloblastoma subtypes. Grade 3 tumor vascularity is also associated with overexpression of the *RNH1, SCG2,* and *AGGF1* genes, and with lower survival rates. Anti-vascularization therapies used in glioblastoma clinical trials may be repurposed for Group 3 medulloblastoma [35]. Ribavirin— antiviral medication for RSV and hepatitis C infections—may have therapeutic potential. Ribavirin significantly prolonged survival and reduced medulloblastoma cell growth in a mouse model of Group 3 medulloblastoma [34].

ABC transporters are implicated in the resistance of medulloblastoma cells to radiation and chemotherapy treatment. In Group 3 subtype mouse models, the ABCG2 transporter is overexpressed. Inhibiting this transporter in in vivo studies enhanced the antiproliferative response to topotecan chemotherapy [59]. ABCA8 and ABCB4 family transporters are highly expressed in the SHH pathway-driven tumors, and confer resistance to radiation therapy [60]. 

### 3.4. Group 4

The Group 4 subtype comprises approximately 35% of medulloblastoma, and occurs in children of all ages (median age, 9 years). There is a 3:1 predominance of males to females in this group. Group 4 medulloblastoma is the least understood medulloblastoma subtype. The Group 4 and 3 subtypes share many features. Like the Group 3 subtype, Group 4 medulloblastoma occurs in the midline vermis, and an underlying molecular pathway does not yet explain its pathogenesis. However, the two non-WNT, non-SHH medulloblastoma subtypes have *MYC* and *OTX2* amplifications [42]. About 6–9% of Group 4 tumors are associated with molecular targets (*KDM6A*, *ZMYM3*, *KTM2C*, and *KBTBD4*) [14]. In preclinical mouse models, aberrant signaling of ERBB4-SRC receptor tyrosine kinase and *TP53* inactivation may induce a tumor resembling Group 4 medulloblastoma [36]. These findings support exploring tyrosine kinase inhibitors such as dasatinib as potential therapeutic agents. The estimated five-year overall survival rate of Group 4 medulloblastoma is 75%. Survival is reduced by metastasis [61]. Low-risk and standard-risk disease is characterized by a loss of chromosome 11 and a lack of metastasis. High-risk disease is defined by metastasis at the time of diagnosis. Interestingly, an ‘intermediate’ type of medulloblastoma, with transitional features of Group 3 and Group 4, has a better prognosis than Group 3 or 4 alone [62].

## 4. Clinical Management of Medulloblastoma

### 4.1. Overview

At diagnosis, an essential prognostic factor in medulloblastoma patients is the anatomical extent of the tumor [61]. Generally, treatment consists of surgical resection followed by chemotherapy and radiation. This therapy results in a survival rate of 70–80% in children with disease confined to the primary site.

### 4.2. Surgical Resection

Historically, the standard of care for medulloblastoma consists of surgery followed by chemotherapy and radiation. Maximal safe surgical resection is advised in pediatric patients with medulloblastoma [28,63]. Surgical procedures are usually performed in the sitting or prone position [64]. Midline tumor growth may obstruct the fourth ventricle and create obstructive hydrocephalus that may require CSF diversion before, during, or after surgical resection through extra ventricular drains, endoscopic third ventriculostomy, or a ventriculoperitoneal shunt [65,66]. While patients benefit significantly from surgical resection, procedures with longer anesthesia duration may be associated with significant neurocognitive decline [67]. 

### 4.3. Radiation Therapy

Radiation therapy (XRT) is a core component of medulloblastoma therapy. In general, children except for infants may receive XRT. Multiple randomized trials suggest that a period of 4 to 5 weeks between surgery and radiation allows wound healing while reducing the risk of tumor regrowth. However, radiation therapy can be delivered at other times between 21 days to 90 days following surgical resection with similar effectiveness [68,69,70,71]. Standard XRT consists of 23.4 Gy to the spine and brain, followed by a boost of 30.6 Gy to the tumor region. Traditionally, the entire cerebellum received XRT. XRT is delivered in 30 to 33 daily fractions of 1.8 Gy for a 54 Gy–59.4 Gy total dose [66]. 

The radiotherapy dose required by pediatric patients with average-risk disease has been evaluated. A Children’s Oncology Group Phase III trial showed that the boost volume for craniospinal irradiation (CSI) could be safely decreased without compromising the patient survival seen after a standard CSI boost to the skull base. Patients aged 3 to 21 years who received the smaller boost had a similar five-year event-free survival (EFS) rate (81%) compared to the standard boost (83%) and the same overall survival rate of 85%. However, the Children’s Oncology Group found reduced survival in children between 3 and 7 years old with average-risk disease receiving a lower boost dose. Low-dose CSI in this population had a 71.4% 5-year event-free survival rate compared to 82.9% for standard-dose CSI. Despite the reduced 5-year survival rate, decreased CSI in these younger patients was associated with better neurocognitive outcomes and a 7-point higher intelligence quotient (IQ) than patients receiving the standard-dose CSI [72].

Often in medulloblastoma, radiation must be delivered to the entire brain and spine to reduce the risk of tumor recurrence. A growing body of literature suggests that proton radiotherapy may treat medulloblastoma better than photon radiotherapy. Traditionally, pediatric medulloblastoma was treated with X-ray photon radiotherapy. Unlike photons, the newer proton therapy uses charged particles whose tissue penetration depth can be controlled. Proton radiotherapy spreads less radiation to healthy tissue than photon radiotherapy, minimizing off-target side effects, [73] reducing psychological, social, and functional adverse effects, and prolonging overall survival [74].

One long-term effect of radiation is diminished cognitive and intellectual abilities. Kallahey, et al., used longitudinal intelligence data from 79 pediatric medulloblastoma patients receiving either proton (PRT) radiotherapy or photon (XRT) radiotherapy. The 79 patients were divided into two groups, and controlled for confounding variables. Patients receiving PRT had more favorable intellectual outcomes than those receiving XRT, including a higher long-term intelligence quotient (IQ) and better working memory, perceptual reasoning, and processing speed [75]. However, another recent phase 2 single-arm study demonstrated that proton radiotherapy was as toxic as traditional photon XRT, and provided no survival advantage in treating pediatric medulloblastoma [76]. In a retrospective study, Zhang, et al., identified a small cohort of pediatric patients who received passively scattered proton CSI or field-in-field photon CSI. They found that patients receiving proton CSI had lower predicted risks of cardiac mortality and secondary cancer incidence than conventional photon CSI [77]. 

### 4.4. Chemotherapy

Chemotherapy is essential to medulloblastoma therapy, targeting residual and micrometastatic disease that cannot be surgically resected or irradicated fully by radiotherapy. Children tolerate chemotherapy better than adults in many types of cancer, including medulloblastoma [78].

One adjuvant chemotherapy regimen for standard-risk pediatric medulloblastoma disease is vincristine, cisplatin, cyclophosphamide, and lomustine [66]. In 2006, the Children’s Oncology Group recommended administering vincristine weekly during radiation therapy, followed by eight cycles of vincristine, cisplatin, and either cyclophosphamide or lomustine [79]. However, vincristine was not considered part of the standard of care for medulloblastoma in a recent clinical trial (SJMB03, NCT00085202) [80].

Using chemotherapy avoids irradiating the developing brain and spinal cord, and radiation-related neurological deficits for infants and young children [81,82,83]. The HIT-2000 trial found that systemic chemotherapy combined with intraventricular methotrexate was a suitable alternative to craniospinal irradiation in children under 4 years of age. These patients demonstrated a 93% 5-year progression-free survival, and 100% overall survival. Of note, over 75% of patients in this study had the SHH subtype [84].

For children older than 3 years with high-risk disease, chemotherapy may be added to radiotherapy to treat surgically unresectable or metastatic tumor. However, studies in recent years indicate that chemotherapy may contribute to toxicity without conferring significant survival benefit. The addition of carboplatin improved 5-year event-free survival for high-risk patients by 19%, although this effect was limited to the Group 3 subtype [85].

### 4.5. Immunotherapy

Clinical trials of immunotherapy are underway in medulloblastoma patients. Immunotherapy has shown efficacy in treating non-CNS tumors [86]. Immunotherapy depends on the patient’s immune system identifying and destroying tumor cells. Tumors resistant to radiotherapy and chemotherapy may be susceptible to immunotherapy because its mechanism of action does not depend on DNA damage, repair, and replication. Additionally, if effective, immunotherapy could spare pediatric patients the cognitive and other neurological side effects of radiotherapy and chemotherapy on the developing central nervous system. Immunotherapy approaches for pediatric medulloblastoma include oncolytic viral therapy and vaccine therapy. 

Oncolytic viral therapy uses an inactivated virus strain to invade and replicate within cancer cells, stimulating the body’s natural immune system to destroy the invaded tumor cells selectively. Several viruses were tested in preclinical models, including HSV-1, measles, reovirus, adenovirus, and parvovirus [87]. Phase I clinical trials in patients with recurrent or refractory medulloblastoma are using poliovirus (*NCT03043391*), measles virus (*NCT02962167*), wild-type reovirus combined with sargramostim (*NCT02444546*), cytomegalovirus RNA-pulsed dendritic cells (*NCT03615404*), and HSV G207, an experimental virus using herpesvirus (*NCT03911388* and *NCT02457845*) [88]. These studies are ongoing and have not reported their findings. 

DNA and RNA-based cancer vaccines deliver genetic information encoding tumor-specific antigens recognized by a patient’s immune system [89]. In contrast, peptide cancer vaccines utilize synthetic peptides to stimulate antigen-presenting cells and tumor-specific T cells [90]. Vaccine therapies for medulloblastoma have had limited success in preclinical and Phase 1 trials. A phase I clinical trial (NCT01171469) is investigating the maximum tolerated dose of a vaccine of autologous dendritic cells reactive to allogeneic brain tumor stem cells. A different phase I trial (NCT01326104) studies a novel vaccine therapy using total tumor RNA-loaded dendritic cells and ex vivo-expanded autologous lymphocytes. This vaccine immunotherapy will be administered concurrently with radiation therapy. The efficacy endpoint of this ongoing trial in pediatric patients with recurrent medulloblastoma is prolonged survival. A phase II clinical trial in patients with recurrent medulloblastoma (NCT00014573) combines chemotherapy with vaccine therapy, stem cell transplantation, and IL-2. In this trial, induction paclitaxel and cyclophosphamide are followed by high-dose cisplatin, cyclophosphamide, and carmustine, and a vaccine developed with autologous tumor cells. 

### 4.6. Imaging-Based Diagnosis

Imaging to diagnose primary disease and detect distant metastases has advanced significantly. MRI remains the imaging mainstay. MRI sequences for diffusion-weighted imaging (DWI) and apparent diffusion coefficient (ADC) maps predict the diagnosis of the various histological subgroups of medulloblastoma [19]. Specifically, large cell/anaplastic group tumors have increased ADC and ring enhancement. Ring enhancement also predicts the extent of tumor necrosis. In addition, MRI can distinguish medulloblastoma from ependymoma, both prevalent posterior fossa tumors in children. Medulloblastoma has significantly higher choline (Cho) levels and choline/N-acetyl aspartate (Cho/NAA) ratios than ependymoma. The optimal Cho/NAA diagnostic threshold predicting medulloblastoma diagnosis is 1.24 [27]. Pineoblastoma is a tumor that shares clinical and histopathological features with medulloblastoma. Mass spectrometric imaging (MSI) uses lipidomic profiles to distinguish pineoblastoma from medulloblastoma [91]. Mouse models have enabled early MRI detection of medulloblastoma metastasis. Three-dimensional matrix-assisted laser desorption/ionization mass spectrometric imaging (MALDI-MSI) identifies typical lipids associated with metastasis [92]. The heterogeneous enhancement of medulloblastoma on MRI is thought to result from the breakdown of the tumor’s blood-brain barrier [93]. 

### 4.7. Quality of Life

Several studies have examined quality of life in medulloblastoma patients. Children with medulloblastoma experience various symptoms that can improve after treatment. Over 60% of medulloblastoma patients have ataxia, which is considerably higher than seen in other childhood tumors [94]. A recent longitudinal study recorded the course of ataxia for two years following surgery for medulloblastoma. Ataxia scores and functional mobility scores improved most significantly in the first 3 months after surgical resection, and gradually after that. 

Cisplatin-induced hearing loss (CIHL) is a well-documented pediatric medulloblastoma treatment complication [95,96,97]. A recent Phase 3 randomized cooperative group trial found that when the antioxidant sodium thiosulfate (STS) was administered with appropriate timing, it protected against CIHL without interfering with cisplatin effectiveness [98].

## 5. Financial Burden of Medulloblastoma Treatment

### 5.1. Treatment Costs for Individual Patients

Treatment for pediatric medulloblastoma often financially burdens affected families. Chemotherapy, radiation, and surgery affect a child’s physical and psychological development, while treatment expenses impact the family’s socioeconomic well-being. The most significant source of costs occurs at the time of diagnosis and in the post-therapy period when adverse effects of treatment arise [99]. In addition to medical expenses, nonmedical costs associated with treatment and the loss of parents’ wages often reduce the family’s yearly gross income by more than 25%. The National Children’s Cancer Society estimates that the average cost per child with cancer is over $800,000 in the United States [77].

### 5.2. Treatment Costs for National Healthcare Systems

The cost and method of pediatric medulloblastoma treatment depend on the patient’s healthcare system, as well as the country in which the patient seeks care. For example, the individual burden of cost is much higher in the United States compared to many European countries. Proton therapy’s cost-benefit ratio has been compared to traditional photon therapy in different countries. Using the incremental cost-effectiveness ratio (ICER) as the primary outcome, Japanese, Canadian, American, and Swedish groups reported that proton beam therapy was more cost-effective than photon therapy in their respective countries due to its lower rate of adverse events [100,101,102,103]. Proton therapy had a lower incidence than photon therapy of hearing loss, reduced intelligence, growth hormone deficiency, and hypothyroidism related to radiation doses delivered to the hypothalamus [104,105]. However, a Brazilian group suggested that proton therapy was not cost-effective if only a limited number of patients at the national level were using it [106]. PBT was cost-effective only if a country treated more than 150 patients with PBT annually. In the event that national health leaders concluded that PBT is more cost-effective than photon XRT for malignancies other than medulloblastoma, then PBT could be cost-effective for medulloblastoma in a national healthcare system like Brazil’s, where the demand for pediatric medulloblastoma treatment remains low. Overall, proton beam therapy has been shown to be a cost-effective strategy for treating pediatric medulloblastoma compared with standard XRT photon radiotherapy. 

## 6. Conclusions and Future Directions

Medulloblastoma is one of the most prevalent and lethal childhood brain tumors. Increasing knowledge over the past decade about the molecular underpinnings and subtypes of medulloblastoma has introduced risk stratification to treatment strategies. This delineation has transformed the landscape of conventional treatment for pediatric medulloblastoma, decreased unnecessary toxicity, and increased overall survival. While surgery, radiotherapy, and chemotherapy have significantly advanced, further work is needed, in order to understand how to better utilize molecular markers in designing future medulloblastoma therapeutic strategies. Further research is also required to improve medulloblastoma treatment effectiveness, reduce treatment morbidity, and preserve patients’ functional and cognitive abilities.

## Figures and Tables

**Figure 1 cancers-14-02285-f001:**
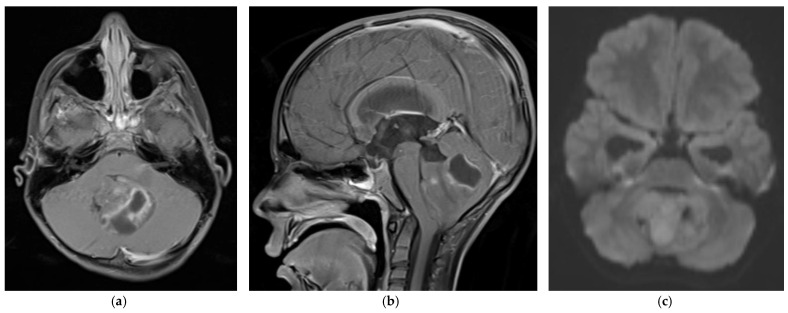
5 year old boy with non-SHH/WNT medulloblastoma. Axial (**a**) and sagittal (**b**) T1-weighted MRI scans after intravenous contrast demonstrate a mass with cystic and solid components that enhances and extends into the fourth ventricle. The axial diffusion-weighted image (DWI) (**c**) shows restricted diffusion (higher signal intensity) within the tumor. The temporal horns of the lateral ventricles are distended, signifying obstructive hydrocephalus.

**Figure 2 cancers-14-02285-f002:**
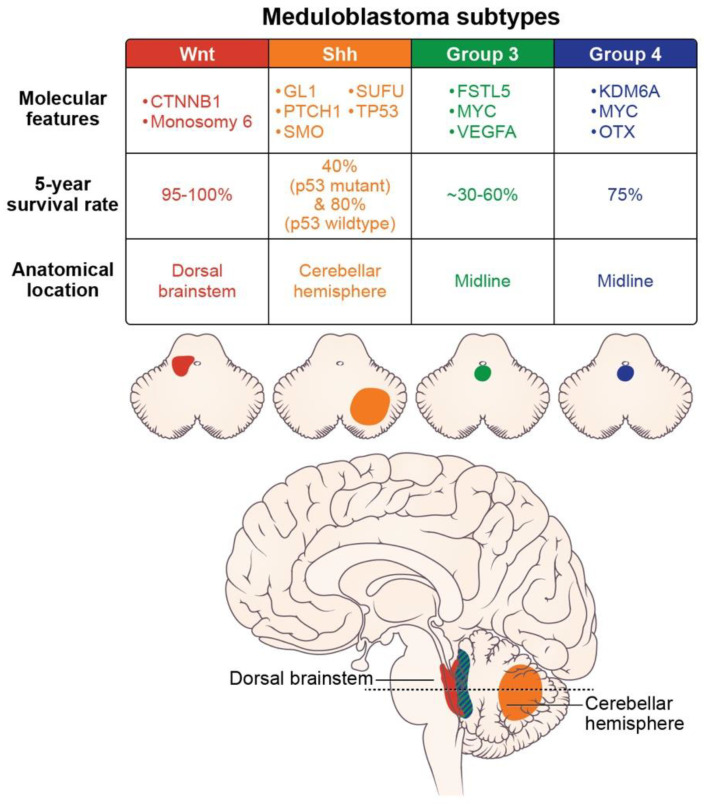
Molecular features (somatic mutations and amplifications), prognosis, and location of the medulloblastoma subtypes (axial images adapted from Juraschka, et al., 2019 [14]).

**Table 1 cancers-14-02285-t001:** Clinical trials and therapies in development for the four pediatric medulloblastoma subtypes.

** *WNT Subtype* **	
Clinical trial: Reducing doses of craniospinal radiation and chemotherapy	NCT01878617: A Clinical and Molecular Risk-Directed Therapy for Newly Diagnosed Medulloblastoma
	NCT02066220: International Society of Paediatric Oncology (SIOP) PNET 5 Medulloblastoma
	NCT02724579: Reduced Craniospinal Radiation Therapy and Chemotherapy in Treating Younger Patients with Newly Diagnosed WNT-Driven Medulloblastoma
Proposed therapy: WNT antagonists	Phoenix, et al. (2016) [31] reported that WNT antagonists block the formation of a blood-brain barrier, and thereby promote chemotherapy penetration and high intratumoral drug concentrations.
** *SHH Subtype* **	
Proposed therapy: nanoparticles	Valcourt, et al. (2020) [32] and Caimano, et al. (2021) [33] reported their development of nanoparticles that encapsulate SMO or GLI inhibitors to improve drug delivery to this tumor subtype.
** *Group 3 Subtype* **	
Proposed therapy: Ribavirin	Huq, et al. (2021) [34] reported therapeutic potential for ribavirin to reduce medulloblastoma cell growth and prolong survival.
Proposed therapy: Anti-vascularization therapy	Thompson, et al. (2017) [35] reported increased vascularity in Group 3 tumors and proposed using anti-VEGFA anti-vascularization therapy to inhibit tumor growth.
** *Group 4 Subtype* **	
Proposed therapy: anti-ERBB4-SRC receptor tyrosine kinase	Forget, et al. (2018) [36] demonstrated that the combination of TP53 inactivation and aberrant signaling of the ERBB4-SRC receptor tyrosine may induce Group 4-like tumor growth. They suggested molecular therapies to inhibit these effects.

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
