# Peer review of "New Developments in the Pathogenesis, Therapeutic Targeting, and Treatment of Pediatric Medulloblastoma"

_cancers, 2022, doi:10.3390/cancers14092285_

Round 1

Reviewer 1 Report

The manuscript focuses on the overview of current diagnosis and treatment methods for CNS tumor class medulloblastoma. This is a useful summary of current status. There are only several minor aspects in the manuscript that would be important to fix.

First, it’s important to note that both SHH and G3/4 MB classes have from WHO 2021 additional classification splitting into subclasses e.g. groups I-VII. This is more research associated block, but should be considered seriously since some clear factors help to improve diagnosis e.g. SHH with P53 LFS syndromes or Group 3/4 II associated with MYC, there would suggest to note this in the introduction and uncover more in detail. 

Additional missing information is the usage of molecular biology techniques (e.g. sequencing, methylation arrays etc) to improve the diagnosis and improve treatment methods selection, even though a separate block about Imaging is present.

Further minor fixes & suggestion:

L55: It’s known that SHH has more clear connection to p53 mutations, while G3 association with MYC amplifications is most verified critical survival factor (PMIDs: 23835706, 31076851)

L62: “trail far behind”: inclusion of reference is critical

L159: Figure 2 legend: unclear what is meant by molecular features, should be noted somatic mutations/amplifications

L217: transcriptomics is combined also with methylation in diagnosis

L223: mouse models noted, but references are missing

L221: another known mechanism of G3 tumor formations is associated with GFI epigenetic changes (PMID: 25043047)

L229: “However, the Group”: what is meant here? Children’s Oncology? Abbreviation would be helpful to explain

L340: “has shown efficacy in treating non-CNS tumors “ : reference?

L373: It’s a bit hard to follow this block since previous segments 4.2-4.5 were related to treatment, while 4.6 focuses on diagnosis. Maybe subblocks structure could be introduced (e.g. Treatment, Diagnosis, etc)? Moreover, molecular biology techniques were not stated as note above, even though became additional useful methods for optimal diagnosis / treatment selection.

L430: dot before reference [108]

Reviewer 2 Report

Thank you for the opportunity to review this interesting and concise review.

I have the following comments:

  1. Line 44 states: "This age distribution highlights the current understanding of these tumors as remnants of aberrant embryonic cerebellar cells."  Please add source/citation here.
  2. I suggest to cut out the following two sentences after that (starting in line 45) completely. Today, prognosis depends primarily on the molecular subtype (as is explained very well in depth later).
  3. I strongly recommend to delete the small passage about the histology of the blood-brain-barrier breakdown in line 390-394. If looked at the quoted study (95) here, only five cases of medulloblastoma were studied with IHC. This is not sufficient for a quality review.
  4. The chapter about the individual financial burden (5.1.) should stress the difference in international health care systems more. In many European Countries the individual burden is not nearly as high, as in the US.
